# EP4 as a Negative Prognostic Factor in Patients with Vulvar Cancer

**DOI:** 10.3390/cancers13061410

**Published:** 2021-03-19

**Authors:** Anna Buchholz, Aurelia Vattai, Sophie Fürst, Theresa Vilsmaier, Christina Kuhn, Elisa Schmoeckel, Doris Mayr, Christian Dannecker, Sven Mahner, Udo Jeschke, Helene H. Heidegger

**Affiliations:** 1Department of Obstetrics and Gynecology, University Hospital, LMU Munich, Marchioninistraße 15, 81377 Munich, Germany; buchholz.anna@web.de (A.B.); aurelia.vattai@med.uni-muenchen.de (A.V.); sophie.fuerst@med.uni-muenchen.de (S.F.); Theresa.Vilsmaier@med.uni-muenchen.de (T.V.); christina.kuhn@uk-augsburg.de (C.K.); christian.dannecker@med.uni-augsburg.de (C.D.); Sven.Mahner@med.uni-muenchen.de (S.M.); Helene.Heidegger@med.uni-muenchen.de (H.H.H.); 2Department of Obstetrics and Gynecology, University Hospital Augsburg, Stenglinstrasse 2, 86156 Augsburg, Germany; 3Department of Pathology, LMU Munich, Thalkirchner Str. 142, 80337 Munich, Germany; elisa.schmoeckel@med.uni-muenchen.de (E.S.); doris.mayr@med.uni-muenchen.de (D.M.)

**Keywords:** EP4, PGE2, COX-2, cancer survival, prognosis, vulvar carcinoma

## Abstract

**Simple Summary:**

Within this study we investigated the expression of EP4 in vulvar cancer, its correlation with clinic-pathological parameters, its association with overall survival, and the effect oft EP4 antagonism on vulvar cancer cells, aiming to find a prognostic and potentially targetable marker in vulvar cancer. Cox regression revealed EP4 as an independent negative prognostic factor for overall survival when other factors were taken into account. We could show in vitro that EP4 antagonism attenuates both viability and proliferation of vulvar cancer cells. Further investigation of the EP4 signaling pathway and its role in the micro tumor environment in vulvar cancer could lead to a deeper understanding of the molecular mechanisms of cancer genesis.

**Abstract:**

New prognostic factors and targeted therapies are urgently needed to improve therapeutic outcomes in vulvar cancer patients and to reduce therapy related morbidity. Previous studies demonstrated the important role of prostaglandin receptors in inflammation and carcinogenesis in a variety of tumor entities. In this study, we aimed to investigate the expression of EP4 in vulvar cancer tissue and its association with clinicopathological data and its prognostic relevance on survival. Immunohistochemistry was performed on tumor specimens of 157 patients with vulvar cancer treated in the Department of Obstetrics and Gynecology, Ludwig-Maximilian-University of Munich, Germany, between 1990 and 2008. The expression of EP4 was analyzed using the well-established semiquantitative immunoreactivity score (IRS) and EP4 expression levels were correlated with clinicopathological data and patients’ survival. To specify the tumor-associated immune cells, immunofluorescence double staining was performed on tissue samples. In vitro experiments including 5-Bromo-2′-Deoxyuridine (BrdU) proliferation assay and 3-(4,5-Dimethylthiazol-2-yl)-2,5-diphenyltetrazoliumbromid (MTT) viability assay were conducted in order to examine the effect of EP4 antagonist L-161,982 on vulvar carcinoma cells. EP4 expression was a common finding in in the analyzed vulvar cancer tissue. EP4 expression correlated significantly with tumor size and FIGO classification and differed significantly between keratinizing vulvar carcinoma and nonkeratinizing carcinoma. Survival analysis showed a significant correlation of high EP4 expression with poorer overall survival (*p* = 0.001) and a trending correlation between high EP4 expression and shorter disease-free survival (*p* = 0.069). Cox regression revealed EP4 as an independent prognostic factor for overall survival when other factors were taken into account. We could show in vitro that EP4 antagonism attenuates both viability and proliferation of vulvar cancer cells. In order to evaluate EP4 as a prognostic marker and possible target for endocrinological therapy, more research is needed on the influence of EP4 in the tumor environment and its impact in vulvar carcinoma.

## 1. Introduction

Vulva carcinoma is a relatively uncommon tumor, representing about 2–5% of all gynecological malignancies [1]. It is mainly a disease in postmenopausal women; however, incidence rates have been increasing in recent years, especially in younger women due to the rise in human papilloma virus (HPV) infections [2]. Squamous cell carcinoma is the most common histological subtype, accounting for 80–90% of all malignancies of the vulva. An estimated 21–43% of vulva carcinomas worldwide are related to human papilloma virus (HPV) infection [3,4,5]. The other HPV-independent group, initiated by p53 mutation, is linked to chronic inflammatory skin diseases such as lichen sclerosis [6,7]. Especially in locally advanced tumor and in lymph node positive patient groups, prognosis remains poor. Predominant treatment is radical surgery, often accompanied by adjuvant or neoadjuvant radiotherapy/chemoradiation. However, radical surgery can lead to reduced quality of life and morbidity [8]. Furthermore, treatment options for advanced vulva cancer are limited due to the lack of large prospective randomized clinical trials regarding systemic treatment [9]. In this context, predictive biomarkers and target-based therapies would play an important role in the improvement of clinical outcomes, especially in advanced stages of disease. In recent times, the search for prognostic markers has gained more and more momentum. The role of cyclooxygenase enzyme 2 (COX-2), successional prostaglandin E2 (PGE2) and its receptors have been investigated in many different tumor entities, such as colon cancer, prostate and lung cancers, as well as in breast cancer and gynecological tumors such as ovarian, cervical and endometrial cancers [10,11,12,13]. PGE2 is known to play an important role in pain, fever, inflammation, mucosal integrity and vascular homeostasis. Moreover, previous studies demonstrated the pivotal impact of PGE2 in carcinogenesis, tumor growth, metastasis and tumor-associated angiogenesis [14,15,16]. PGE2 is synthesized from arachidonic acid through NF-kB inducible cyclooxygenase enzyme-2 (COX-2) and mediates its effects through its specific ligands, prostaglandin E2 receptor 1–4 (EP 1–4). Prostaglandin E2 receptors are G-protein coupled receptors activating different signal pathways, including the cAMP/PKA/CREB pathway, the Ras/Raf/MEK/ERK pathway and the GRK/β-arrestin/Src/PI3K/GSK3/b-catenin pathway, which subsequently stimulate the transcription of target genes, including cyclin-D1, c-myc and VEGF [17,18]. As demonstrated in previous studies, COX-2 inhibitors, such as nonsteroidal anti-inflammatory drugs (NSAIDs) or celecoxib, might be effective in cancer prevention and therapy [19]. However, the possibility of severe cardiovascular side effects limits their application in cancer treatment [20]. Therefore, targeting COX-2 or PGE2 indirectly by their downstream ligands (EP1-4) could be a promising approach.

The objective of this study was to shed light on the role of EP4 receptor expression in vulvar cancer. We investigated the expression of EP4 in vulvar cancer, its correlation with clinicopathological parameters, its association with overall survival, and the effect oft EP4 antagonism on vulvar cancer cells, aiming to find a prognostic and potentially targetable marker in vulvar cancer.

## 2. Results

Of all specimens, we obtained successful EP4 staining from 131 patients. We could find EP4 cytosolic staining (IRS ≥ 1) in 93.9% (123/131 cases) of cases, with a median IRS of 6 in 9.2% of cases. High expression (IRS 9–12) was found in 26.7% of specimens, compared to moderate expression (IRS 6–8) in 26.8%, weak expression (IRS 3–4) in 26% and no expression (IRS 0–2) in 20.6%. In comparison to vulvar cancer specimens, benign tissue of the vulva showed no EP4 expression. Regarding the histological subtype, 91.7% of the specimens were keratinizing squamous cell carcinomas, 5.1% were nonkeratinizing squamous cell carcinomas, 1.9% were verrucous squamous cell carcinomas and 1.3% were warty squamous cell carcinomas. Clinicopathological characteristics are displayed in Table 1.

### 2.1. Correlation between EP4-Positive Staining and Clinicopathological Parameters

We examined the correlation between positive EP4 staining and clinicopathological parameters using Spearmen’s test. EP4 did not correlate with p16 status (*p* = 0.174), tumor grading (*p* = 0.252), primary lymph node metastasis (*p* = 0.357), or the number of tumor foci (*p* = 0.944). However, a significant correlation was observed between positive EP4 staining and greater tumor size (pT) (*p* < 0.001) and EP4 and high FIGO classification (*p* = 0.003) (Figure 1). Kruskal–Wallis tests showed significant differences among different FIGO stages (*p* = 0.014) and pT (*p* = 0.002). In addition, Mann–Whitney U test revealed that EP4 expression was higher in keratinizing squamous cell carcinoma than nonkeratinizing squamous cell carcinoma (*p* = 0.024).

### 2.2. Role of EP4 for Survival

For three patients no survival follow-up could be obtained. After the observation period, 51 of 157 patients were still alive. In total, 103 of 157 patients died during the observation period; median follow-up time was 4.79 years (SD = 6.17).

EP4 positivity was associated with poorer prognosis in overall and disease-free survival. As depicted in the Kaplan–Meier curves, enhanced expression of EP4 (IRS ≥ 3) in vulva cancer patients significantly correlated with shorter overall survival (median estimate 13.7 years vs. 7.2 years; *p* = 0.001) and showed a trend for a correlation with shorter disease-free survival (median estimate 13.7 years vs. 9.7 years; *p* = 0.069) time after diagnosis (Figure 2). 

### 2.3. Cox Regression of EP4 IRS with Clinicopathological Variables

As described above (methods, statistics), a Cox regression analysis was performed to ascertain the prognostic relevance of EP4 expression when other prognosticators were taken into account. Enhanced EP4 receptor positivity (IRS ≥ 3) was found to be an independent prognostic factor for poorer overall survival (*p* = 0.016, hazard ratio (HR) = 2.924, 95% confidence interval (CI)—1.225–6.980) and disease-free survival (*p* = 0.045, hazard ratio (HR) = 2.683, 95% confidence interval (CI)—1.020–7.054). Furthermore, age at diagnosis was an independent prognostic factor associated with both, disease-free and overall survival. Moreover, tumor grade and lymph node status at the time of operation were prognostic factors for overall survival. Data of the multivariate analysis is displayed in Table 2.

### 2.4. EP4 Positive Tumor—Associated Immune Cells

In total, 36.6% of the specimens showed EP4 receptor-positive tumor-associated immune cells. These immune cells were identified as macrophages using immunofluorescence double staining with anti-EP4 and anti-CD68 antibodies (Figure 3). EP4 positivity in macrophages did not correlate with clinicopathological parameters. In the subgroup of specimens with EP4-negative macrophages, patients with a tumor IRS ≥ 3 had a significant worse outcome in terms of disease-free survival (13.2 years vs. 8.3 years; *p* = 0.29). This effect did not show in the subgroup of specimens with EP4-positive macrophages. EP4 status in macrophages itself had no significant impact on patients’ survival.

### 2.5. EP4 Antagonist Reduced the Proliferation and Viability of Vulvar Cancer Cells

EP4 expression was detected in both cell lines. However, EP4 expression was higher in A-431 cells in comparison to SW-954 cells. β-actin was used as loading control. Benign human epidermal tissue showed insignificant expression of EP4 (Appendix A).

Stimulation of vulvar cancer cells with increasing concentrations of L-161,982 for 72 h led to decreased viability and proliferation of both A-431 and SW-954 cell lines. A-431 proliferation decreased significantly after stimulation with 10µM (Z = −3.238; *p* = 0.001; *n* = 15) and 100 µM (Z = −3.408; *p* = 0.001; *n* = 15). SW-954 proliferation also decreased significantly after stimulation with concentrations of 10 µM (Z = −2.472; *p* = 0.013; *n* = 15) and 100 µM (Z = −3.411; *p* = 0.001; *n* = 15) (Figure 4). The effect on proliferation was stronger in SW-954, as its EP4 expression is lower and therefore receptors might be saturated faster with L-161,982. Viability decreased significantly after stimulation with 100 µM L-161,982 in both cell lines (Z = −3.408; *p* = 0.001; *n* = 15) (Figure 4) and in SW-954 at a concentration of 10 µM (Z = −2.045, *p* = 0.041; *n* = 15).

## 3. Discussion

Within this study, we observed that EP4 receptor positivity is a frequent finding in vulvar carcinoma, which is the first time EP4 expression and its association with clinicopathological parameters was investigated in vulvar carcinoma. In line with studies showing the impact of COX-2 on carcinogenesis, previous studies also described the effects of EP4 on carcinogenesis in several gynecological cancer models such as breast cancer [21,22], endometrial cancer [23,24], ovarian cancer [25] and cervical cancer [26]. A study by Reader et al. revealed higher expression of EP4 in uterine leiomyosarcoma in comparison to smooth muscle tumors and normal myometrium. It also showed a strongly increased sensitization to docetaxel in sarcoma cells after pretreatment with EP4 antagonist [27]. However, there are only few studies that investigated the role of the COX-2/PGE2/EP receptor axis in vulvar neoplasia. The one study investigating COX-2 expression in vulvar carcinoma reported higher COX-2 expression levels in vulvar neoplasia than in healthy vulvar tissue [28].

In this present study, high EP4 receptor expression correlated with both shorter overall survival and shorter disease-free survival. This is consistent with the study of Fons et al., who reported an association between strong COX-2 expression in vulvar carcinoma and shorter disease-free survival [29]. Moreover, Cox regression analysis revealed EP4 expression, but also age, grading and lymph node metastasis (pN) as independent prognostic factors for overall survival. Our results compare well with findings of previous studies that have proven the importance of lymph node metastasis (pN) and age as prognostic factors in vulvar cancer [30].

Furthermore, we found correlations between higher EP4 expression and FIGO classification, greater tumor size (pT) and the amount of keratin of the tumor. This is in line with an investigation describing significant correlations of higher tumor/stroma COX-2 expression ratio with metastatic lymph node involvement and higher FIGO stage of the tumor [31]. Similar to our results, they detected a trending association between stromal invasion (pT) and COX-2 expression. However, in our analysis EP4 was not associated with lymph node metastasis and grading.

Additionally, we found that EP4 was also detectible in tumor-associated immune cells, which were specified as macrophages by immunofluorescence. Previous investigations identified EP4 as the predominant prostaglandin receptor isoform in human macrophages in cell culture and atheroma tissue [32]. Macrophages are known to be very adaptable cells promoting heterogenic effects depending on their changing surrounding microenvironments [33]. They are known to foster tumor progression by different mechanisms including suppression of antitumor immunity and promotion of angiogenesis and cell invasiveness [34]. In addition to its well-known proinflammatory effect, Takayama et al. showed that endogenous PGE_2_ may act as an anti-inflammatory by suppressing macrophage-derived chemokine production via the EP4 receptor [32]. In our study, the visible trend for a reduction in disease-free survival in the entire patient collective for specimens with tumor EP4 IRS ≥ 3 was statistically significant in the subgroup of patients with EP4-negative tumor-associated macrophages. Nevertheless, in our study, EP4 positivity in tumor-associated macrophages itself did not correlate with reduced disease-free or overall survival. Moreover, we did not find any correlations with other clinicopathological parameters. Therefore, immunohistochemical analysis in our study provides a preliminary insight into EP4 expression in tumor-associated macrophages and further investigations are needed to understand how EP4 in macrophages modulates cancerogenic mechanisms.

As described earlier, HPV infection plays an important role in carcinogenesis of vulvar carcinoma. HPV encodes for oncogenic proteins such as E7 inactivating tumor suppressor p53, E6 inducing degradation of pRb and E5, whose oncogenic mechanism has been studied the least. HPV E5 is located in the endoplasmatic reticulum and is described as playing a supporting role for the viral oncoproteins E6 and E7 by trafficking of cytoplasmic membrane proteins, increased epidermal growth factor receptor (EGFR) signaling and activation of the MAPK pathway [35]. Oh et al. studied the effects of HPV E5 protein on PGE2 signaling in cervical cancer cells and described an induction of EP4 by HPV E5 with epidermal growth factor receptor, COX-2, PGE2, EP2 and EP4, protein kinase A, CREB and CRE being engaged in this induction [36]. However, in our analysis, we could not find correlations between EP4 expression and p16 status in vulvar carcinoma. Multiple studies could verify p16 as a reliable surrogate for HPV-association in cervical and oropharyngeal tumors [37,38,39,40]. However, p16 overexpression was also detected in cases of HPV-independent tumors—this indicates that, in addition to the viral oncogenes E6/7, there must be HPV-independent pathways inducing p16 overexpression [41,42]. Therefore, more studies are needed to examine the accuracy of p16 as a marker for HPV association in vulvar carcinoma and its relationship with the overexpression of EP4. Furthermore, there might be different definitions of p16 positivity in immunohistochemistry [43]. Exclusively strong cytoplasmic and nuclear “block like” staining throughout the whole tumor on slide should be considered as p16-positive [44].

EP4 promotes its tumorigenic effect through several processes in the microtumor environment. Inter alia, previous studies identified PDL-1 expression as being a negative prognosticator in squamous cell carcinoma of the vulva and showed that the number of tumor infiltrating lymphocytes was associated with PD-L1 expression [44]. According to Wang et al., stimulation of PGE2/EP4 signaling pathways resulted in an increased expression of PD-1 in infiltrating CD8+ T-cells in patients with lung cancer [45]. Similar results have been published Miao et. al, who revealed that the combined blockade of PGE_2_ and PD-1 pathways led to a substantially renovated function of cytotoxic lymphocytes [46]. These results indicate that targeting EP receptors in combination with PD-L1 blockade might be a promising diagnostic or therapeutic option for cancer patients in the future.

We could show in vitro that EP4 antagonism can decrease viability and proliferation in vulvar carcinoma cells, supporting previous studies describing the impact of EP4 in carcinogenesis in other tumor entities [12]. This effect on proliferation was stronger in SW-954, as its EP4 expression is lower and therefore receptors might be saturated faster with L-161,982. Parida et al. showed a diminished tumor viability and proliferation of cervical cancer in vitro as well as in vivo after EP4 antagonism [26]. Additionally, in breast cancer cell lines decreased migration and proliferation was described following a treatment with EP4 antagonists [22]. In a study with colon carcinoma cell lines, L-161,982 blocked ERK phosphorylation and thereby inhibited proliferation via the Ras-Raf-MEK-ERK pathway. This effect was the strongest in EP4 regulated pathways compared to the other EP receptors [47].

## 4. Materials and Methods

### 4.1. Patients

The analyzed study group consisted of 177 patients who underwent surgery at the Department of Gynecology and Obstetrics in Munich between 1990 and 2008. Of these 177 samples, 157 were available for immunohistochemical staining. As negative and positive controls for immunohistochemical staining protocols, we used placenta tissue, also obtained from the Department of Gynecology and Obstetrics in Munich. Patients’ median age was 69.5 years (range 20–96 years) and overall median survival was 7.03 years. Immediately after resection, the vulva cancer tissue was fixated in formalin solution and embedded in paraffin.

### 4.2. Immunohistochemistry

The paraffin embedded slides were stained immunohistochemically, as previously described in [48].

First, the slides were dewaxed with xylol for 20 min, washed in alcohol, then incubated in methanol with 3% H_2_O_2_ for 20 min in order to inhibit the endogen peroxidases and were finally rehydrated in descending alcohol. In order to unmask the antigen after formalin-fixation-associated protein-agglomeration, the slides were heated in sodium citrate buffer (pH = 6.0) (0.1 M citric acid and 0.1 M sodium citrate in distilled water) using a pressure cooker. After cooling and washing in PBS, we used blocking solution (Reagent 1, Zytochem-Plus HRP Polymer-Kit (mouse/rabbit) Zytomed Systems Berlin, Germany) for 20 min to avoid nonspecific binding of the primary antibodies. We incubated the slides with the primary anti-EP4 antibody (polyclonal rabbit IgG, HPA012756, Sigma Aldrich, St.Louis, MO, USA) for 16 h, before detecting it with the polymer method via secondary complex (ZytoChem Plus HRP Polymer System mouse/rabbit,) and the chromogen diaminobenzidine (Dako, Hamburg, Germany).

The expression of EP4 in the specimens was analyzed with a Leitz (Wetzlar, Germany) microscope using the well-established semiquantitative immunoreactivity score (IRS). The IRS was derived by multiplying the intensity of the staining (0 = no, 1 = weak, 2 = moderate, 3 = strong staining) with the percentage of stained cells (0 = no staining, 1 ≤ 10% positive cells, 2 = 11–50% positive cells, 3 ≥ 50% positive cells) [49]. Samples with an IRS of 0, 1 or 2 were classified as EP4-negative, and samples with an IRS of 3 or higher were counted as EP4-positive. Eventually, both groups were compared for clinicopathological parameters, progression-free and overall survival. Tumor-associated immune cells were counted per field of view (25× lens) and dichotomized in positive (> 4 per field of view) and negative (≤ 4 per field of view) immune cell staining. Mean values of infiltrating immune cells detected in three different spots of the same individual were calculated [50]. As negative and positive controls, we used placenta tissue, also obtained from the Department of Gynecology and Obstetrics in Munich.

P16 staining was performed on a Ventana Benchmark XT autostainer (Ventana Medical Systems, Oro Valley, AZ, USA) using the p16–primary antibody (clone E6H4E6H4/p16^Ink4a^, Ventana, ready-to-use) and the XT UltraView diaminobenzidine kit (Vector Laboratories, Burlingame, CA, USA) followed by hematoxylin counterstaining (Vector Laboratories). Samples were classified as p16-positive when strong cytoplasmic and nuclear staining was observed throughout the whole tumor on slide (“block” staining). Cases showing a weak or patchy staining were classified p16-negative.

### 4.3. Immunofluorescence

To clearly identify the tumor-associated immune cells, immunofluorescence double staining for EP4 and the macrophage marker CD68 was performed. The tissue slides were preprocessed as they were for immunohistochemistry, as described above. The next steps included the application of blocking solution (UltraVision Protein Block; Thermo Fisher Scientific, Waltham, MA, USA)

The slides were pretreated as they were for immunohistochemistry. To prevent unspecific binding of the primary antibody, a blocking solution (UltraVision Protein Block; Thermo Fisher Scientific) was applied to the slides for 15 min. The slides were incubated for 16 h with a mixed solution of the primary antibodies. After washing the slides thoroughly with PBS, fluorophore-labeled secondary antibodies were applied for 30 min in the dark at room temperature. Finally, the slides were covered with mounting medium (Vectashield H-1200; Vector Laboratories) containing 4′,6-diamidino-2-phenylindole (DAPI) for nuclear counterstaining. All double stainings were observed at 20×, 40× and 63× magnifications using a confocal laser microscope (Axiophot fluorescent microscope; Zeiss, Oberkochen, Germany) and analyzed with the corresponding software AxioVision.

### 4.4. Cell Culture

The human carcinoma cell lines A-431 and SW-954 used in our study as vulva carcinoma models were purchased from the American Type Culture Collection (Manassas, VA, USA). The cells were cultured in Dulbecco’s Modified Eagle’s Medium (DMEM) (Biochrom, Berlin, Germany) containing 10% fetal bovine serum (FCS, Thermo Fisher Scientific) without antibiotic in a humidified incubator at 37 °C with 5% CO_2_ saturation. In preparation for each experiment, cells were counted using a Neubauer cell chamber.

### 4.5. Inhibition with EP4 Antagonist L-161,982

A-431 and SW-954 cells were seeded into 96-well plates at a density of 1 × 10^4^ cells/100 µL/well in quintuplicates and incubated over night for 24 h. The next day, 100 µL of fresh medium containing different concentrations (2, 20 or 200 µM) of EP4 antagonist L-161,982 (Tocris Bioscience, Bristol, UK) or the vehicle DMSO as a control were added to each well, resulting in a final concentration of 1, 10 or 100µM. Afterwards, cells were incubated for 72 h.

### 4.6. Viability Assay

After stimulation with the EP4 antagonist L-161,982 for 72 h, MTT assay was used to examine its effects on the viability of the cells. To each well, 20 μg 3-(4,5-dimethylthiazol-2-yl)-2,5-diphenyltetrazoliumbromid (MTT) (Sigma-Aldrich) solution (5 mg/mL in phosphate-buffered saline PBS) was then added and incubated for 1.5 h at 37 °C. The culture medium along with MTT was then removed. A total of 200 μL DMSO/well was added and mixed thoroughly on the shaker for 5 min at room temperature to dissolve the formazan crystals. Finally, optical density was quantified at 595 nm using an Elx800 universal Microplate Reader. The experiment was repeated three times in order to ensure reliability.

### 4.7. Proliferation Assay

After stimulation with the EP4 antagonist L-161,982 for 72 h, 5-bromo-2’-deoxyuridine (BrdU) incorporation assay (Roche, Basel, Swizzerland) was used to determine the cell proliferation according to the manufacturer’s protocols. For this purpose, BrdU was added in a concentration of 10 µM/well, and the cells were reincubated for 24 h. The medium was removed from the cells, and 200 µL of Fix-Denat was added to each well and incubated for 30 min at room temperature. The supernatant was removed and 100 µL of Anti-BrdU-POD solution was added to the cells for 90 min, binding the BrdU, which was incorporated before in the newly synthesized DNA. In the next step, the wells were carefully washed three times with PBS and subsequently the cells were incubated with a substrate solution (tetramethyl-benzidine) in the dark at room temperature for about 20 min until an increasing blue color became visible. The reaction was stopped by adding 25 µL of 1 M sulfuric acid. Finally, optical density was quantified at 450 nm using an Elx800 universal Microplate Reader. The experiment was repeated three times in order to ensure reliability.

### 4.8. Western Blot

First, cell lysates of 5 × 10^6^ cells per cell line (SW-954 and A-431) were produced using a buffer containing 1 mL of RIPA buffer (Sigma Aldrich, R0278-50ML), 2 µL protease Inhibitor cocktail (Sigma Aldrich, P8340) and 10 µL Natrium-Vanadate. Human keratinozytes were obtained from the Department of Dermatology, University Hospital Augsburg (gift of Prof. Dr. J. Welzel). Lysates of benign human epidermal tissue in liquid nitrogen were produced by additionally using an ultrasonic sonifier cell disruptor B15 (Branson, Brookfield, CT, USA). Proteins were separated by SDS-PAGE electrophoresis and transferred to PVDF membranes. After 2 h incubation in a blocking solution containing TBS/Tween/milk powder, the membranes were incubated at room temperature for 16 h overnight with the primary antibody dilutions polyclonal rabbit anti-EP4 (1:300; ab217966, Abcam, Cambridge, US) and monoclonal mouse anti-β-actin (1:1000, Sigma Aldrich). The next day, the membranes were washed and incubated for 1 h at room temperature with the corresponding 1:1000 dilution of alkaline phosphatase-conjugated secondary antibodies. Staining was carried out using 5-bromo-4-chloro-3-indolyle phosphate/nitroblue-tetrazolium chloride (Promega, Madison, WI, USA) in 0.1-M Tris–HCl, 0.15 M NaCl, pH 9.5. Western blot images were scanned and quantified with Quantity One 4.6.7 (Bio-Rad, Hercules, USA) (Appendix A). Staining intensity of the Western blots was normalized to β-actin staining and is presented in absolute number of densitometry analysis.

### 4.9. Statistics

Data analysis was performed with the Statistical Product and Service Solutions 25 (PASW Statistic, SPSS Inc., IBM, Chicago, IL, USA). Spearmen’s test was used to test for correlations between immunohistochemically staining and clinicopathological parameters. Nonparametric tests (Mann–Whitney U and Kruskal–Wallis tests) were used for group comparisons regarding the IRS of the prostaglandin receptors between independent clinical and pathological subgroups and are displayed as boxplot graphs. A Wilcoxon test was used for the evaluation of viability/proliferation levels between vehicle and antagonist groups. Survival times were analyzed by Kaplan–Meier curves and log-rank testing (Mantel Cox). Cut-off points were acquired by the receiver operator curve (ROC). We considered *p* values ≤ 0.05 as statistically significant.

## 5. Conclusions

In conclusion, we observed that EP4 is an independent prognostic factor for the overall survival in vulvar carcinoma. In addition, the immunohistochemical staining of EP4 is correlated to FIGO classification and tumor size pT, whereas EP4 positivity seems to be independent from p16 status. We could show in vitro that EP4 antagonism attenuates viability and proliferation of vulvar cancer cells. Further investigation of the EP4 signaling pathway and its role in the micro tumor environment in vulvar cancer could lead to a deeper understanding of the molecular mechanisms of cancer genesis.

## Figures and Tables

**Figure 1 cancers-13-01410-f001:**
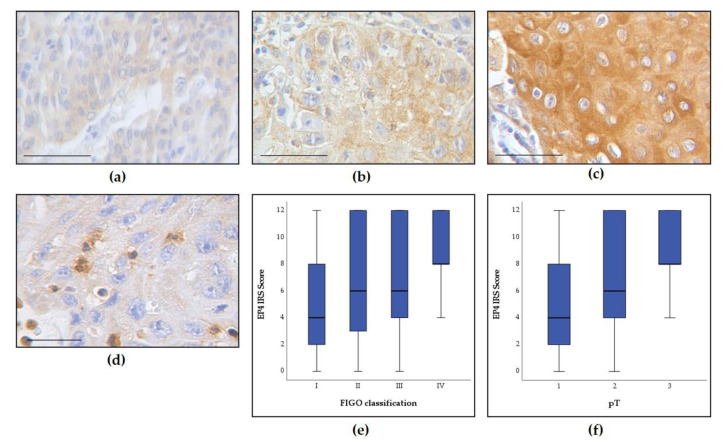
EP4 expression depending on tumor size and FIGO status. Immunohistochemical staining of EP4 in (**a**) vulvar cancer with weak expression (immunoreactivity score (IRS) 4), (**b**) vulvar cancer with moderate expression (IRS 6) and (**c**) vulvar cancer with strong expression (IRS 12). (**d**) Vulvar cancer with tumor-associated macrophages. (**e**,**f**) EP4 expression correlates with FIGO classification (*p* = 0.014) and tumor size (*p* < 0.001). Magnification and scale bars (**a**–**c**) x 25 with scale bar representing 100 µm and (**d**) x 40 with scale bar representing 50 µm.

**Figure 2 cancers-13-01410-f002:**
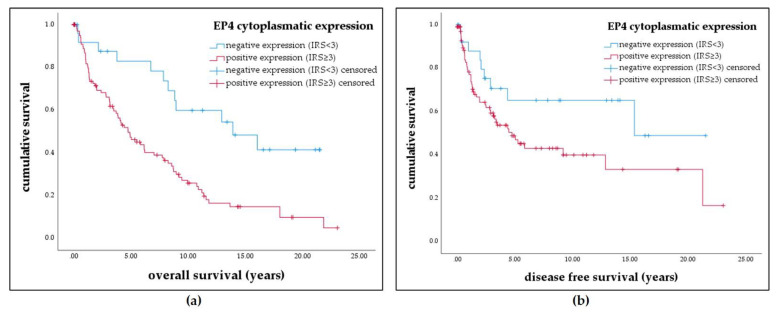
Kaplan–Meier estimates show that overall survival/disease-free survival is worse in patients with high EP4 expression. (**a**) Kaplan–Meier survival univariate analysis for the status cytoplasmatic IRS ≥ 3. Positive expression of EP4 in the cytoplasm significantly reduced overall survival (*p* = 0.001) (**b**) Kaplan–Meier survival univariate analysis for the status cytoplasmatic IRS ≥ 3. Positive expression of EP4 lead to decreased disease-free survival (*p* = 0.069).

**Figure 3 cancers-13-01410-f003:**
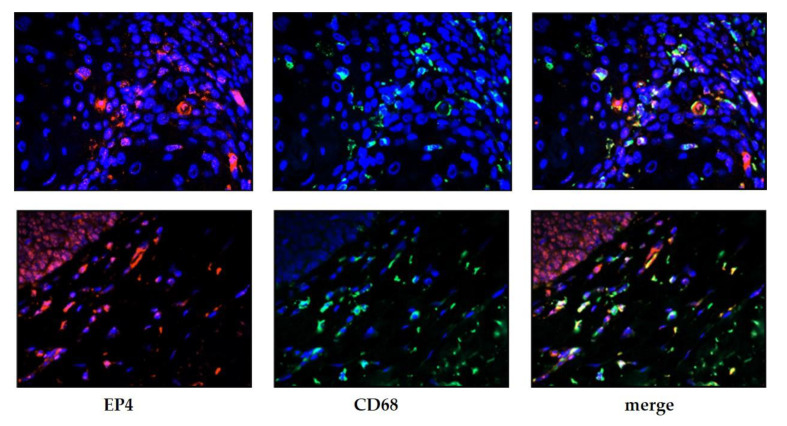
Characterization of immune cells as macrophages using immunofluorescence double staining. Cell nuclei were marked by 4′,6-diamidino-2-phenylindole (DAPI) staining (blue): left column shows EP4-positive cells (red), center column shows CD68-positive cells (green) and right column shows the coexpression of both markers CD68 and EP4.

**Figure 4 cancers-13-01410-f004:**
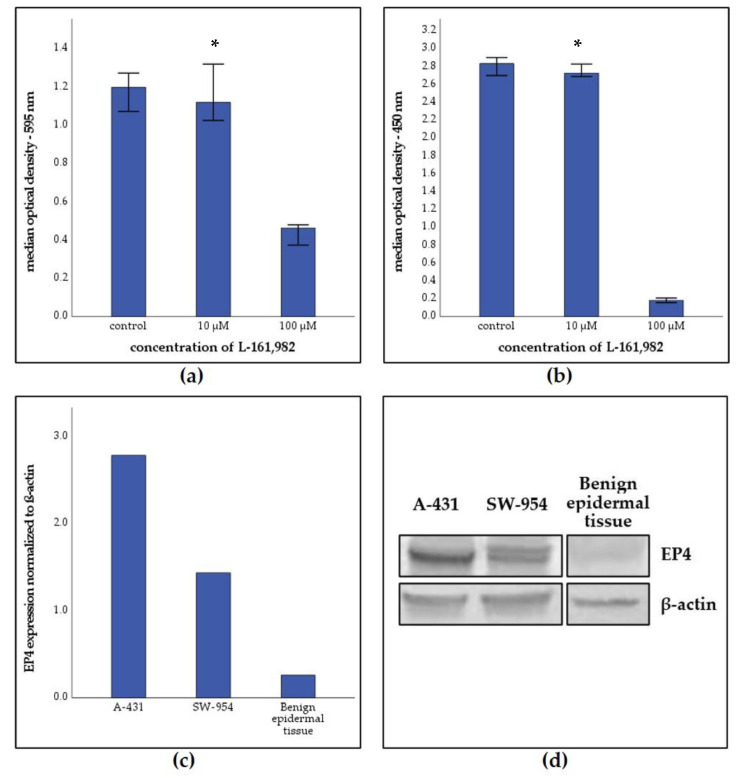
Incubation of vulvar carcinoma cells with L-161,982. (**a**) 3-(4,5-Dimethylthiazol-2-yl)-2,5-diphenyltetrazoliumbromid (MTT) viability assay of A-431 cells after inhibition with different concentrations of L-161,982 for 72 h. * *p* = 0.001. (**b**) 5-Bromo-2’-Deoxyuridine (BrdU) proliferation assay of SW-954 calls after inhibition with different concentrations of L-161,982 for 72 h. * *p* = 0,001. (**c**,**d**) Western blot analysis of EP4 expression in A-431 and SW-954 cell lines and of benign epidermal tissue. (**a**,**b**) Error bars represent 95% confidence intervals. Wilcoxon test was used for the evaluation of viability/proliferation levels between vehicle and antagonist groups.

**Table 1 cancers-13-01410-t001:** Clinicopathological characteristics of the analyzed vulvar carcinoma samples.

Clinicopathologic Parameters		*n*	Percentage (%)
Histology			
	Keratinizing	160	90.4
	Warty/basaloid	17	9.6
Tumor size			
	T1	69	39
	T2	92	52
	T3	9	5.1
	missing	7	3.9
Nodal status			
	N0	78	44.1
	N1	38	21.5
	N2	12	6.8
	missing	49	27.6
Metastasis			
	M0	8	4.5
	missing	169	95.5
FIGO			
	I	61	34.4
	II	54	30.5
	III	47	26.6
	IV	9	5.1
	missing	6	3.4
Grading			
	G1	29	16.4
	G2	108	61
	G3	39	22
	missing	1	0.6
P16 status			
	Positive	38	21.5
	Negative	57	32.2
	missing	82	46.3

**Table 2 cancers-13-01410-t002:** Cox regression of clinicopathological variables regarding overall survival.

Co-Variants	Significance	Hazard Ratio of Exp (B)	Lower 95% CI of Exp (B)	Upper 95% CI of Exp (B)
EP4 IRS ≥ 3	**0.016**	2.924	1.225	6.980
Age	**<0.001**	1.052	1.026	1.080
Grading	**0.007**	1.904	1.197	3.027
pN	**0.033**	1.941	1.055	3.572
pT	0.417	1.506	0.561	4.039
FIGO	0.704	0.863	0.405	1.841
Focalty	0.688	0.843	0.366	1.940
Histology	0.877	1.092	0.360	3.307

Bold number characterize *p*-values below 0.05.

## Data Availability

The data presented in this study are available on request from the corresponding author. The data are not publicly available due to ethical issues.

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
