# Peer review of "EP4 as a Negative Prognostic Factor in Patients with Vulvar Cancer"

_cancers, 2021, doi:10.3390/cancers13061410_

Round 1
Reviewer 1 Report
The authors examined the expression of EP4 in 131 vulvar cancer specimens. They found weak moderate and strong EP4-expression in a third of cases, each. There was a correlation of EP4- expression in tumor cells to tumor size and patient survival. No correlation to p16 and other clinicopathological parameters was detected. Additionally a correlation of EP4 –expression on macrophages with survival is described.
There are some serious issues in this paper:
The EP4-staining in Figure 1a and 1b is weak and poor visible. The authors should provide clear images in 20x or 40x magnification with strong moderate and weak EP4 staining, each. In Figure 1b the macrophages are not detectable. The magnification should be at least 40x to show these cells in appropriate manner. The figures should be replaced.
The FIGO stages 1,00 – 4,00 in Figure 1c do not exist. The authors should use the correct designation for FIGO stage on x axis.
In the material and methods section the authors describe the procedure for immunoflourescence staining of macrophages. There are no criteria to differentiate between positive and negative staining results. The authors should use similar scores as described for immunohistochemistry for detailed examination of EP4 expression of tumor-infiltrating macrophages. In Figure 1b the authors show EP4-expression on macrophages with conventional immunohistochemical methods. Are there special reasons to use immunoflourescence for further investigation? In the paper the EP4 expression was correlated to clinicopathological parameters but for EP4 expression on macrophages not. It could be interesting to provide these data. Wrong spelling “4.3. Immuflourescence”
The new WHO classification of female genital tract tumors from 2020 strongly differentiate between HPV-dependent and HPV-independent vulvar squamous cell carcinomas. HPV is an important factor for tumor development in at least 25% of vulvar cancers. However, the remaining tumors are HPV-negative and have an completely different molecular status as well as clinical course and survival including therapeutic options The marker p16 may be an good surrogate marker for HPV in tumour tissues. There is no information in the method section 4.2. how the p16 staining was carried out and which criteria were used to classify the tumors in p16 positive and negative cancers. The author should provide this data. Furthermore, the molecular examination of tumor tissue to clarify the HPV status is nowadays a routine method either with PCR or RNA-ISH. The value of the present analysis could be improved by use of these routine methods.
Author Response
REVIEWER 1
The authors examined the expression of EP4 in 131 vulvar cancer specimens. They found weak moderate and strong EP4-expression in a third of cases, each. There was a correlation of EP4- expression in tumor cells to tumor size and patient survival. No correlation to p16 and other clinicopathological parameters was detected. Additionally a correlation of EP4 –expression on macrophages with survival is described.
There are some serious issues in this paper:
- The EP4-staining in Figure 1a and 1b is weak and poor visible. The authors should provide clear images in 20x or 40x magnification with strong moderate and weak EP4 staining, each. In Figure 1b the macrophages are not detectable. The magnification should be at least 40x to show these cells in appropriate manner. The figures should be replaced.
Authors’ response:
Thank you for these observations. We replaced the figures by pictures with larger magnifications and better quality. Now in macrophages should be clearly visible (Figure 1e and 1f).
- The FIGO stages 1,00 – 4,00 in Figure 1c do not exist. The authors should use the correct designation for FIGO stage on x axis.
Authors’ response:
We agree and corrected the error (Figure 1.)
- In the material and methods section the authors describe the procedure for immunofluorescence staining of macrophages. There are no criteria to differentiate between positive and negative staining results. The authors should use similar scores as described for immunohistochemistry for detailed examination of EP4 expression of tumor-infiltrating macrophages. In Figure 1b the authors show EP4-expression on macrophages with conventional immunohistochemical methods. Are there special reasons to use immunofluorescence for further investigation?
Authors’ response:
Immunohistochemically, all specimens were analyzed for EP4 positive tumor infiltrating immune cells. To the methods section we added: “Tumor associated immune cells were counted per field of view (25× lens) and dichotomized in positive (> 4 per field of view) and negative (≤ 4 per field of view) immune cell staining. Mean values of infiltrating immune cells detected in three different spots of the same individual were calculated.” (line 278-281) Immunofluorescence was used, to specify the tumor infiltrating immune cells more precisely. Double staining with the CD68 marker for macrophages identified them as macrophages.
- In the paper the EP4 expression was correlated to clinicopathological parameters but for EP4 expression on macrophages not. It could be interesting to provide these data.
Authors’ response:
We analyzed the data using the Spearmen’s test but did not find any significant correlations. We added the negative results with the sentences “EP4 positivity in macrophages did not correlate with clinicopathological parameters.” (line 135-136) and. “EP4 status in macrophages itself had no significant impact on patients’ survival.” (line 138-140) to the results section. However, in the subgroup of specimens with EP4 negative macrophages, survival analysis showed that patients with a tumor IRS ≥ 3 had a significant worse outcome in disease-free survival (13.2 years vs. 8.3 years; p = 0.29). (line 136-138)
- Wrong spelling “4.3. Immuflourescence”
Authors’ response:
Thank you for pointing out the spelling error. We replaced it by “Immunofluorescence” (line 292).
- The new WHO classification of female genital tract tumors from 2020 strongly differentiate between HPV-dependent and HPV-independent vulvar squamous cell carcinomas. HPV is an important factor for tumor development in at least 25% of vulvar cancers. However, the remaining tumors are HPV-negative and have a completely different molecular status as well as clinical course and survival including therapeutic options The marker p16 may be an good surrogate marker for HPV in tumor tissues. There is no information in the method section 4.2. how the p16 staining was carried out and which criteria were used to classify the tumors in p16 positive and negative cancers. The author should provide this data.
Authors’ response:
We revised the methods section to include this information: “P16 staining was performed on a Ventana Benchmark XT autostainer (Ventana Medical Systems, Oro Valley, AZ) using the p16 – primary antibody(clone E6H4E6H4/p16Ink4a, Ventana, ready-to-use) and the XT UltraView diaminobenzidine kit (Vector Laboratories, Burlingame, CA) followed by hematoxylin counterstaining (Vector Laboratories, Burlingame, CA). Samples were classified as p16 positive, when strong cytoplasmic and nuclear staining was observed throughout the whole tumor on slide (“block” staining). Cases showing a weak or patchy staining were classified p16-negative.” (lines 184–190)
- Furthermore, the molecular examination of tumor tissue to clarify the HPV status is nowadays a routine method either with PCR or RNA-ISH. The value of the present analysis could be improved by use of these routine methods.
Authors’ response:
We agree, using molecular methods as PCR would improve the reliability of the study regarding its standardized feasibility. However, due to the retrospective design of our study, we could only use paraffine embedded tissue for our analyses. We will consider this point in the planning of upcoming prospective studies.
Reviewer 2 Report
In this paper, Buchholz and colleagues examined the role of PGE2 EP4 receptor in vulvar carcinoma through immunohistochemical analysis of tumor tissue for EP4 expression and antagonism of EP4 through the use of small molecule inhibitors on vulvar carcinoma cell lines. The authors reported that EP4 expression correlated with decreased overall survival and disease free survival. Lastly, inhibition of EP4 with small molecule antagonist resulted in reduction in proliferation and viability of in vitro of vulvar carcinoma cell lines.
Immunohistochemical analysis: Information is missing on what controls were used for analysis of EP4 expression. Also, there is no information in regards to EP4 staining on normal vulvar tissue in order to compare to what is observed in the cancer tissue. Please make Figure 2 bigger as it is difficult to read the text.
Immunofluorescence analysis: It would be beneficial to see an example with EP4 negative macrophages in addition to the EP4 positive macrophage tissue staining for Figure 3.
Western blot analysis: The authors state that there is higher expression of EP4 expression but quantification by densitometry is unclear. A normal or benign vulva cell line should be used in order to compare the cancer cell line expression with a normal cell line and the loading control should be used to normalize the expression but not to determine fold change of expression of EP4. This should be done relative to a normal or benign control cell line. Figure 4 in the figure legend stimulation is not the correct term to use being as this is an EP4 antagonist and thus the cells should be inhibited not stimulated. What is the effect of this antagonist on a normal vulvar cell line in comparison to the cancer cell line given the high concentration of antagonist used (100 uM). There are no errors bars for any of the graphs – error bars should be present if experiments were replicated as indicated. Also there is no statistical analysis information for Figure 4.
The discussion is missing references in regards to uterine leiomyosarcoma and EP4 expression. Tumor microenvironment instead of micro tumor environment. In the discussion, in regards to immune cells and EP4, there is no integration of the author’s data with this discussion, as the authors reported that EP4 negative macrophages in EP4 positive tumor samples was associated with worse outcome in disease free survival. The authors fail to address these data in the discussion. In the last paragraph in the discussion there are no references to support the statement “that EP4 antagonism can decrease viability and proliferation in vulvar carcinoma cells, supporting previous studies describing the impact of EP4 in carcinogenesis in other tumor entities.”
Author Response
REVIEWER 2
In this paper, Buchholz and colleagues examined the role of PGE2 EP4 receptor in vulvar carcinoma through immunohistochemical analysis of tumor tissue for EP4 expression and antagonism of EP4 through the use of small molecule inhibitors on vulvar carcinoma cell lines. The authors reported that EP4 expression correlated with decreased overall survival and disease free survival. Lastly, inhibition of EP4 with small molecule antagonist resulted in reduction in proliferation and viability of in vitro of vulvar carcinoma cell lines.
- Immunohistochemical analysis: Information is missing on what controls were used for analysis of EP4 expression. Also, there is no information in regards to EP4 staining on normal vulvar tissue in order to compare to what is observed in the cancer tissue.
Authors’ response:
As negative and positive control, we used Placenta tissue, also obtained from the Department of Gynecology and Obstetrics in Munich (line 181-183).
We agree and added a picture of EP4 staining on benign vulvar tissue in figure 1a. and added the sentence “In comparison to vulvar cancer specimens, benign tissue of the vulva showed no EP4 expression.” (line 83-84) in the results section.
- Please make Figure 2 bigger as it is difficult to read the text.
Authors’ response:
Thank you for the notation, we enlarged Figure 2.
- Immunofluorescence analysis: It would be beneficial to see an example with EP4 negative macrophages in addition to the EP4 positive macrophage tissue staining for Figure 3.
Authors’ response:
We would like to apologize that, due to supply difficulties, we are not able to provide an additional negative example.
- Western blot analysis: The authors state that there is higher expression of EP4 expression but quantification by densitometry is unclear. A normal or benign vulva cell line should be used in order to compare the cancer cell line expression with a normal cell line and the loading control should be used to normalize the expression but not to determine fold change of expression of EP4. This should be done relative to a normal or benign control cell line. Figure 4 in the figure legend stimulation is not the correct term to use being as this is an EP4 antagonist and thus the cells should be inhibited not stimulated. What is the effect of this antagonist on a normal vulvar cell line in comparison to the cancer cell line given the high concentration of antagonist used (100 uM).
Authors’ response:
We agree, it would be interesting to compare the effects of EP4 antagonists on healthy cells. We will take up this point for future studies. Unfortunately, until now it is not possible to obtain a benign vulvar cell line. In our study, western blot was mainly conducted for quantification of the expression in the different cell lines in order to find suitable cell lines for the study. Figure 4c shows results from quantification with Quantity One 4.6.7 software (Bio-Rad, Munich, Germany) in absolute numbers, after being normalized to ß-actin expression which served as loading control.
We changed the legend of figure 4 and replaced stimulation by inhibition. (line 163 and 164)
- There are no errors bars for any of the graphs – error bars should be present if experiments were replicated as indicated.
Authors’ response:
We agree and added error bars to the diagrams (Figure 4.) (line 166).
- Also there is no statistical analysis information for Figure 4.
Authors’ response:
Statistical information can be found in section 4.9 Statistics: “Wilcoxon test was used for the evaluation of viability/proliferation levels between vehicle and antagonist groups.” (line 367-368)
- The discussion is missing references in regards to uterine leiomyosarcoma and EP4 expression.
Authors’ response:
Thank you for the remark. We added to the discussion: “A study by Reader et al. revealed higher expression of EP4 in uterine leiomyosarcoma in comparison to smooth muscle tumors and normal myometrium. It also showed a strongly increased sensitization to docetaxel in sarcoma cells after pre-treatment with EP4 antagonists (Reader, J., et al., EP4 and Class III β-Tubulin Expression in Uterine Smooth Muscle Tumors: Implications for Prognosis and Treatment. Cancers (Basel), 2019. 11(10).) ” (line 173–176)
- Tumor microenvironment instead of micro tumor environment.
Authors’ response:
Thank you for the correction. We replaced it by “micro tumor environment”. (line 386)
- In the discussion, in regards to immune cells and EP4, there is no integration of the author’s data with this discussion, as the authors reported that EP4 negative macrophages in EP4 positive tumor samples was associated with worse outcome in disease free survival. The authors fail to address these data in the discussion.
Authors’ response:
We took up your valuable point and added the following passage to the discussion:
“Additionally, we found, that EP4 was also detectible in tumor associated immune cells, which were specified as macrophages by immunofluorescence. Previous investigations identified EP4 as the predominant prostaglandin receptor isoform in human macrophages in cell culture and atheroma tissue [32]. Macrophages are known to be very adaptable cells promoting heterogenic effects depending on its changing sur-rounding microenvironment [33]. They are known to foster tumor progression by different mechanisms including suppression of anti-tumor immunity and promotion of angiogenesis and cell invasiveness [34]. Besides its well-known pro-inflammatory effect, Takayama et all showed that endogenous PGE2 may act anti-inflammatory by suppressing macrophage-derived chemokine production via the EP4 receptor [32]. In our study, the visible trend for a reduction of disease-free survival in the entire patient collective for specimens with tumor EP4 IRS ≥ 3, was statistically significant in the subgroup of patients with EP4-negative tumor associated macrophages. Nevertheless, in our study, EP4 positivity in tumor associated macrophages itself did not correlate with reduced disease-free or overall survival. Moreover, we did not find any correlations with other clinicopathological parameters. Therefore, immunohistochemical analysis in our study provides a preliminary insight in EP4 expression in tumor associated macrophages and further investigations are needed to understand how EP4 in macrophages modulates cancerogenic mechanisms.” (line 196-213)
- In the last paragraph in the discussion there are no references to support the statement “that EP4 antagonism can decrease viability and proliferation in vulvar carcinoma cells, supporting previous studies describing the impact of EP4 in carcinogenesis in other tumor entities.”
Authors’ response:
We agree and added the reference “Ye, Y., et al., COX-2-PGE2-EPs in gynecological cancers. Arch Gynecol Obstet, 2020.” to the discussion. (line 238)
Round 2
Reviewer 1 Report
The authors give an response to the review of their manuscript “EP4 as a Negative Prognostic Factor in Patients with Vulvar Cancer”.
Issue 1: The figures 1e and 1f show the same. Figure 1e should be deleted. Figure 1a shows nothing relevant for this analysis and can also be deleted.
Issue 2: In general the FIGO use roman numbers in their classification.
Issue 3: An explanation about the reason of choosing the magic number 4 would be valuable.
Issue 4: acceptable.
Issue 5: acceptable.
Issue 6: acceptable.
Issue 7: The authors write in the abstract, line 16 “… clinicopathological data including HPV status …”. The HPV status of vulvar carcinomas is not analyzed in this study. Therefore, authors should accept this fact and delete this statement in the abstract. Furthermore, in the discussion section the authors write “… EP4 expression is independent from the HPV surrogate p16 status …”. The authors should state that p16 expression as studied in this analyze is not suited to say something about the relationship between EP expression and HPV association of the tumours, because the big overlap between p16 expression in HPV associated tumors (by far not always block like) and HPV independent tumors (often with substantial p16 expression )
Author Response
REVIEW 1
Issue 1: The figures 1e and 1f show the same. Figure 1e should be deleted. Figure 1a shows nothing relevant for this analysis and can also be deleted.
Authors’ response:
We agree and deleted the redundant figures. (Figure1. Line 99-103)
Issue 2: In general the FIGO use roman numbers in their classification.
Authors’ response:
We changed the Figure, it now contains roman numbers (Figure 1e.)
Issue 3: An explanation about the reason of choosing the magic number 4 would be valuable.
Authors’ response:
This cut off value is established in our lab and has been previously described by Badmann, S., et al. We added the citation “Badmann, S., et al., M2 Macrophages Infiltrating Epithelial Ovarian Cancer Express MDR1: A Feature That May Account for the Poor Prognosis. Cells, 2020. 9(5).” to the methods section. (Line 315)
Issue 4: acceptable.
Issue 5: acceptable.
Issue 6: acceptable.
Issue 7: The authors write in the abstract, line 16 “… clinicopathological data including HPV status …”. The HPV status of vulvar carcinomas is not analyzed in this study. Therefore, authors should accept this fact and delete this statement in the abstract. Furthermore, in the discussion section the authors write “… EP4 expression is independent from the HPV surrogate p16 status …”. The authors should state that p16 expression as studied in this analyze is not suited to say something about the relationship between EP expression and HPV association of the tumours, because the big overlap between p16 expression in HPV associated tumors (by far not always block like) and HPV independent tumors (often with substantial p16 expression )
Authors’ response:
We agree and deleted the statement from the abstract (Line 16). Additionally, we added the following section to the manuscript:
“Multiple studies could verify p16 as a reliable surrogate for HPV-association in cervical and oropharyngeal tumors [37-40]. However, p16 overex-pression was also detected in cases of HPV independent tumors- This indicates that, besides via the viral oncogenes E6/7, there must be HPV independent pathways in-ducing p16 overexpression [41, 42]. Therefore, more studies are needed to examine the accuracy of p16 as a marker for HPV-association and in vulvar carcinoma and its rela-tionship with the overexpression of EP4. Furthermore, there might be different defini-tions of p16 positivity in immunohistochemistry [43] Exclusively strong cytoplasmic and nuclear “block like” staining throughout the whole tumor on slide should be con-sidered p16 positive [44].” (Line 251- 259)
Reviewer 2 Report
Follow up to the initial review for this paper, the authors responded to most of my issues but failed to respond to one key issue.
1. Correction accepted.
2. Correction accepted.
3. Explanation accepted.
4.
- Western blot analysis: The authors state that there is higher expression of EP4 expression but quantification by densitometry is unclear. A normal or benign vulva cell line should be used in order to compare the cancer cell line expression with a normal cell line and the loading control should be used to normalize the expression but not to determine fold change of expression of EP4. This should be done relative to a normal or benign control cell line. Figure 4 in the figure legend stimulation is not the correct term to use being as this is an EP4 antagonist and thus the cells should be inhibited not stimulated. What is the effect of this antagonist on a normal vulvar cell line in comparison to the cancer cell line given the high concentration of antagonist used (100 uM).
Authors’ response:
We agree, it would be interesting to compare the effects of EP4 antagonists on healthy cells. We will take up this point for future studies. Unfortunately, until now it is not possible to obtain a benign vulvar cell line. In our study, western blot was mainly conducted for quantification of the expression in the different cell lines in order to find suitable cell lines for the study. Figure 4c shows results from quantification with Quantity One 4.6.7 software (Bio-Rad, Munich, Germany) in absolute numbers, after being normalized to ß-actin expression which served as loading control.
We changed the legend of figure 4 and replaced stimulation by inhibition. (line 163 and 164).
In multiple publications that utilize this cell line since the 1990s, primary human keratinocytes are used as a control. Multiple types of primary are available at ATCC. I recommend acquiring primary human keratinocytes and testing EP4 antagonist on these cells.
5. Correction accepted.
6. Please add statistical information to the figure legend in addition to the materials and methods.
7. Correction accepted.
8. Correction accepted.
9. Correction accepted.
10. Correction accepted.
Author Response
REVIEW 2
We agree, it would be interesting to compare the effects of EP4 antagonists on healthy cells. We will take up this point for future studies. Unfortunately, until now it is not possible to obtain a benign vulvar cell line. In our study, western blot was mainly conducted for quantification of the expression in the different cell lines in order to find suitable cell lines for the study. Figure 4c shows results from quantification with Quantity One 4.6.7 software (Bio-Rad, Munich, Germany) in absolute numbers, after being normalized to ß-actin expression which served as loading control.
We changed the legend of figure 4 and replaced stimulation by inhibition. (line 163 and 164).
In multiple publications that utilize this cell line since the 1990s, primary human keratinocytes are used as a control. Multiple types of primary are available at ATCC. I recommend acquiring primary human keratinocytes and testing EP4 antagonist on these cells.
Authors’ response:
We conducted a western blot analysis on lysates obtained of benign human epidermal tissue (keratinocytes are the predominant cell type in human epidermis). Lysates of benign human epidermal tissue in liquid nitrogen were produced by additionally using an ultrasonic sonifier cell disruptor B15 (Branson, Brookfield, CT, USA) and continued the experiment as indicates in the methods section. As the western blot analysis showed very low expression of EP4, we could not conduct MTT and BrDU assays on human keratinocytes. We completed the methods and results section (Line 147 and Line 185 -187).
- Correction accepted.
- Please add statistical information to the figure legend in addition to the materials and methods.
Authors’ response:
We added the statistical information to the figure legend. (Figure 4, Line 191 – 192)
- Correction accepted.
- Correction accepted.
- Correction accepted.
- Correction accepted.
Round 3
Reviewer 2 Report
Updates are acceptable except please change stimulation in material and methods when referring to use of EP4 antagonist. It's not stimulation, it's inhibition.